# The Influence of Microbiota on Breast Cancer: A Review

**DOI:** 10.3390/cancers16203468

**Published:** 2024-10-13

**Authors:** Cara-Xenia-Rafaela Neagoe, Mihaela Ionică, Octavian Constantin Neagoe, Adrian Pavel Trifa

**Affiliations:** 1Doctoral School, “Victor Babes” University of Medicine and Pharmacy, Eftimie Murgu Square No. 2, 300041 Timisoara, Romania; cara.neagoe@umft.ro; 2Second Clinic of General Surgery and Surgical Oncology, Emergency Clinical Municipal Hospital, 300079 Timișoara, Romania; ionica.mihaela@umft.ro; 3Second Discipline of Surgical Semiology, First Department of Surgery, “Victor Babeș” University of Medicine and Pharmacy, 300041 Timișoara, Romania; 4Breast Surgery Research Center, “Victor Babeș” University of Medicine and Pharmacy, 300079 Timișoara, Romania; 5The Discipline of Genetics, “Victor Babes” University of Medicine and Pharmacy, 300041 Timisoara, Romania; adrian.trifa@umft.ro; 6Department of Genetics, Clinical Hospital of Infectious Diseases and Pneumophthisiology “Dr. Victor Babes” Timisoara, 300041 Timisoara, Romania; 7Center for Research and Innovation in Personalized Medicine of Respiratory Diseases, “Victor Babeș” University of Medicine and Pharmacy, 300041 Timișoara, Romania

**Keywords:** breast cancer, microbiota, tumor immune microenvironment, tumor microenvironment, microbiome, microbial signatures

## Abstract

**Simple Summary:**

Breast cancer continues to represent a leading cause of death among women globally. Recent studies have shown a growing link between breast cancer and the living microorganisms from the tumoral breast tissue, known as microbiota. This review looks at how the microbiota is found in healthy breast tissue and the changes observed during cancer development and progression. The microbiota has been shown to affect cancer growth, spread, and resistance to treatments by interacting with the tumor’s environment. Moreover, this review explores how different breast cancer types have distinct microbial profiles. Future studies could improve breast cancer care by endowing the microbiota as a diagnostic, prediction and treatment marker.

**Abstract:**

Breast cancer remains one of the leading causes of death among women worldwide, and recent research highlights its growing connection to alterations in the microbiota. This review delves into the intricate relationship between microbiotas and breast cancer, exploring its presence in healthy breast tissue, its changes during cancer progression, and its considerable impact on both the tumor microenvironment (TME) and the tumor immune microenvironment (TIME). We extensively analyze how the microbiota influences cancer growth, invasion, metastasis, resistance to drugs, and the evasion of the immune system, with a special focus on its effects on the TIME. Furthermore, we investigate distinct microbial profiles associated with the four primary molecular subtypes of breast cancer, examining how the microbiota in tumor tissues compares with that in adjacent normal tissues. Emerging studies suggest that microbiotas could serve as valuable diagnostic and prognostic biomarkers, as well as targets for therapy. This review emphasizes the urgent need for further research to improve strategies for breast cancer prevention, diagnosis, and treatment. By offering a detailed examination of the microbiota’s critical role in breast cancer, this review aims to foster the development of novel microbiota-based approaches for managing the disease.

## 1. Introduction

Breast cancer is of great interest to many physicians, surgeons, and pathologists because it has the highest prevalence among the site-specific and gender-specific cancers in women. Recent data has shown that breast cancer malignancy exceeds that of lung cancer regarding incidence in 2020; therefore, it is the main cause of cancer worldwide [1].

The majority of malignant breast tumors are adenocarcinomas, and their origin is in the terminal duct lobular unit (TDLU). Carcinoma of the breast is classified as ductal or lobular based on the anatomical and histological structure from which the malignancy arises [2]. When referring to tumoral behavior, both ductal and lobular carcinomas can be defined as in situ carcinoma or invasive carcinoma. Invasive carcinomas have metastatic potential, and they encompass invasive breast carcinoma of no special type (IDC in the past), invasive lobular carcinoma (ILC), tubular carcinoma, mucinous carcinoma, invasive micropapillary carcinoma, and many more [3]. Invasive breast carcinoma of no special type (IDC-NST) is the most frequent form of invasive breast cancer (70–80% of all invasive breast cancers), followed by ILC (which accounts for 10% of all invasive malignancies of the breast) [2,4,5]. 

The molecular classification of breast cancer and more specifically the immunohistochemical-based substitute molecular classification has been supported because it offers guidance to the physician in selecting the best treatment. It is a major prognostic and predictive role in the tumoral response to hormonal therapy [6]. ER, PR, HER2, and Ki-67 proteins are IHC markers that are used for the identification of the following molecular subtypes of breast cancer: luminal A (ER+, PR ≥ 20%, HER2−, Ki-67 low), luminal B (ER+, PR < 20% and/ or HER2+ and/or Ki-67 high), HER2-overexpression (ER−, PR−, HER2+), and triple-negative breast cancer (ER−, PR−, HER2−). The most usual molecular subtype of breast cancer is represented by the luminal A subtype (which accounts for 40–50% of all invasive breast cancers) and has the best outcome among all molecular subtypes [5,7,8].

A variety of risk factors are associated with the development of breast cancer, such as the early onset of puberty, early menarche, late age at first pregnancy, lack of breastfeeding, late onset of menopause, familial history of breast malignancy (linked to pathogenic variants especially in high-penetrance genes, most frequently *BRCA1*, *BRCA2*, and *PALB2*), alcohol consumption, obesity, dietary fat intake, hormone replacement therapy, the intake of contraceptives containing estrogen and progesterone, and environmental factors [9,10,11,12,13]. All the risk factors mentioned above are well-known and thoroughly studied; though, not all cases of breast malignancies can be explained through these factors. 

Recent studies have highlighted a connection between the local microbiota of the breast and the occurrence of malignant breast tumors, suggesting that the population of bacteria colonizing the mammary gland might represent a new risk factor for breast cancer. Previously, it was thought that breast tissue had no microbial colonization, but now it is understood that the mammary gland has a unique microbiome. The rich adipose tissue, the abundant blood supply, and lymphatic drainage of the breast makes it a perfect environment for the proliferation of microorganisms [14,15,16]. 

In the same line, an association has been observed between the specific microbial pattern of the four breast cancer subtypes and the outcome of the malignant disease [17]. Furthermore, the intratumoral microbiota has a crucial role in promoting the spreading and the endurance of tumor cells, encouraging metastatic colonization of different organs in breast cancer [18,19]. The changes and the imbalance occurring in the local microbiota when breast cancer develops were also analyzed. It was found that a certain type of bacteria (*Sphingomonas yanoikuyae*) is predominant in normal breast tissue (surrounding the tumoral tissue). Another type of bacteria (*Methylobacterium radiotolerans*) is most prevalent in the breast tissue with malignant transformation. The dysbiosis in the microbiota is explained by the fact that the amount of *S. yanoikuyae* decreases considerably in breast cancer while the amount of *M. radiotolerans* remains the same [20]. These findings emphasize the importance of the microbiome and microbiota as potential diagnostic markers for malignant breast disease.

All the aspects mentioned above confirm that the microbiota has a significant role in breast cancer regarding risk, diagnosis, therapy, and outcome. It is of vital importance to research and comprehend new associations between microbiotas and breast cancer in order to improve diagnostic and therapeutic algorithms. The aim of this review is to emphasize the involvement of microbiotas in the aforementioned aspects of breast cancer. 

Key findings: The microbiota has a significant role in breast cancer, regarding risk, diagnosis, therapy, and outcome;The microbiota modulates the TME and TIME, which have a major influence on malignant cell viability, cancer proliferation, and invasion;Breast tissue has a unique microbiota (in terms of healthy or tumoral tissue), and shifts in bacterial taxa and metabolic reprogramming of the microbiota are preliminary events in malignancy development;The composition of the breast tissue resident microbiota changes dramatically as the breast tissue begins to suffer malignant transformation, holding specific microbial patterns;Breast-cancer-tissue-associated microbiotas may provide the role of a biomarker or therapeutic target in the future.

## 2. Materials and Methods 

Literature research was conducted using PubMed and Google Scholar databases from 2014 onwards. The keywords used for database interrogations were as follows: microbiota; breast cancer; tumor microenvironment (TME); bacterial taxa; metabolic reprogramming of microbiota; host–microbiota interactions; microbial signatures; microbial dysbiosis; cancer proliferation; cancer metastasis; drug resistance; immune escape; diagnostic biomarkers; prognostic biomarkers; therapeutic targets; immunotherapy. A total of 6439 publications were found. The search results were reviewed, with the inclusion criteria of studies placing emphasis on breast cancer tissue resident microbiotas, microbial dysbiosis in breast cancer, specific microbial signatures in breast cancer, a correlation between breast microbiotas and cancer development, and the potential of microbiotas as diagnostic and prognostic biomarkers in breast cancer. Three of the authors independently selected potential articles for inclusion based on the title, with further refinement of the selection process based on abstract review. Following the PRISMA guidelines, after the exclusion of duplicates, the abstract review for suitability, and finally the full-text analysis of the remaining papers, a total number of 35 articles were selected for inclusion in the present review, as can be observed in the flow diagram (Figure 1). This study was not registered.

## 3. The Connection between Microbiota and Cancer

### 3.1. Relationship between Microbiota and Human Health

The interactions between microbiotas and human organisms are utterly important for the maturation of the immune system. It is acknowledged that any imbalance in the microbiota can lead to conditions mediated by an abnormal host immune response: allergic and autoimmune disorders, and metabolic and chronic inflammatory diseases, including various cancer type [21]. The influence of dysbiosis in the microbiota towards immune-mediated disorders, including cancer, surpasses the organs commonly colonized by extensive microbial communities (such as gastrointestinal tract, skin, mucosal tissue, and the respiratory tract) and includes conditions localized in non-barrier tissues or with less microbial colonization [21,22,23].

Gut microbiotas have significant roles in the human body, influencing the maturation of the immune system and cellular homeostasis, modulating neurologic signaling and host–cell multiplication, maintaining a barrier against pathogens, regulating the endocrine functions of the gut, participating in the synthesis or degradation of many substances (for instance, the synthesis of vitamin K2, the metabolism of bile salts, dietary constituents, medications, etc.) [24,25,26,27,28]. Disturbances in the human host–microbiota relationship in the gastrointestinal tract can lead to exaggerated immune responses against the gut microbiota, promoting the pathogenesis of IBD (inflammatory bowel disease), and the imbalance in the gut microbiota between commensal microorganisms and opportunistic pathogens is associated with colorectal cancer (CRC) [21,29]. These findings highlight a strong relationship between the microbiota and human health.

### 3.2. Impact of the Microbiota on the Tumor Microenvironment (TME) and Tumor Immune Microenvironment (TIME)

The tumor microenvironment (TME) has a major influence on malignant cell viability, cancer proliferation, and invasion [30]. The TME incorporates cellular and non-cellular components. The cellular components of TME are cancer cells, cells of the innate immune system (e.g., neutrophils, macrophages, NK cells, dendritic, and myeloid-derived suppressor cells), lymphocytes B and T, stromal cells (e.g., fibroblasts and adipocytes), and endothelial cells [30,31]. The extracellular matrix (ECM) represents a non-cellular constituent of the TME, which integrates molecules like collagen, laminin, hyaluronan, fibronectin, and elastin. The TME also includes soluble factors (e.g., cytokines, hormones, chemokines, growth factors, matrix remodeling enzymes, and mediators of inflammation), which enable cellular interactions. Other ways of intercellular communication in the TME include exosomes, apoptotic bodies, circulating tumor cells, and cell-free DNA [30,31,32,33,34]. The TME is shaped by the microbiota-derived metabolites, and the intercellular communication within TME is influenced by the microbiota, as it modulates the main functions in carcinogenesis: inflammation, proliferation, angiogenesis, the initiation of the epithelial–mesenchymal transition, invasion, metastasis, and immune escape [30,35]. In a study conducted by Xuan et al., the composition of the microbiota in malignant tissue and in paired normal tissue adjacent to the tumor within samples from 20 patients diagnosed with ER-positive breast cancer was analyzed and compared (both qualitatively and quantitatively). Next-generation sequencing techniques were used in order to complete a high-resolution scrutiny of the microbiota in tumor and paired normal tissue samples [20]. From a qualitative perspective, *Sphingomonas yanoikuyae* prevailed in paired normal tissue, as it was detected in 95% of samples. On the other hand, *Methylobacterium radiotolerans* prevailed (as it was detected in 100% of samples) in the malignant breast tissue [20,30]. From a quantitative perspective, the levels of *S. yanoikuyae* were considerably elevated in the paired normal tissue, as it was not found in the tumor tissue samples. *M. radiotolerans* was found in both the paired normal and malignant tissue samples, but between the absolute levels in these types of samples, a significant difference was not demonstrated. In the tumor tissue samples, the relative predominance of *M. radiotolerans* is explained by a reduction in other bacterial species. In the malignant tissue, compared to the paired normal tissue, the level of *S. yanoikuyae* decreases notably as the quantity of *M. radiotolerans* remains at a constant value. An obvious connection between the TME and microbiota is highlighted by the fact that the bacterium *Sphingomonas yanoikuyae* expresses certain glycosphingolipid ligands for invariant natural killer T cells. The glycosphingolipids from *S. yanoikuyae* induce the activation of invariant natural killer T cells, as these lymphocytes have important roles in tumor rejection and immunological surveillance [20,30,36].

The tumor immune microenvironment (TIME) encompasses a variety of cellular components and substances, such as malignant cells, immune cells, enzymes, and cytokines. These components are either anti-tumor factors or pro-tumor factors. Tumors have the ability to escape from immune surveillance by promoting an immunosuppressive tumor microenvironment and encouraging cancer progression. The anti-tumor immune response is driven mainly by T lymphocytes, such as cytotoxic T lymphocytes (CTL) and T helper cells [37]. By understanding the complex process of eluding the immune system by tumors, various antibody-based immunotherapies have been discovered in order to reset the TIME and undermine malignant cells. For instance, immune checkpoint inhibitors act by blocking CTLA-4 and PD-1/PD-L1, thereby diminishing the functional inhibition of T lymphocytes [37,38,39]. It was discovered that triple-negative breast cancer (TNBC) expresses an increased level of PD-L1, leading to the use of a PD-1 immune checkpoint blockade for treating TNBC [40,41]. Pembrolizumab was used in addition to neoadjuvant chemotherapy and also as an adjuvant treatment after performing breast surgery. Pembrolizumab-based therapy was applied to patients with incipient cancer, improving progression free survival and increasing the pathological complete response rate to more than 60% [40,42]. 

The TIME is strongly influenced by systemic patient-dependent factors such as age, gender, dietary habits, physical activity, adiposity, and the composition of microbiota. The inflammatory state of the patient can also shape the nature of the TIME in premalignant lesions; therefore, two scenarios are possible: supporting cancer proliferation and progression or promoting tumoral clearance by the immune system [38]. 

Certain mechanisms can describe the relationship between the microbiota and TIME [43]. One hypothesis is that microbial antigens can imitate tumor antigens. The microbial antigens can be presented to immune cells and tumor cells, and this process can activate a response from the immune system, as effector T cells can recognize and destroy the antigen-presenting cell. As a result of antigen similarity, T lymphocytes can identify malignant cells that present similar antigenic determinants and can eliminate them [43,44,45]. Another hypothesis is that microbial residents can interact with pattern recognition receptors (PRRs) and so modulate the TIME [43,46].

### 3.3. The Link between Microbiotas and Cancer Proliferation, Invasion, Metastasis, Drug Resistance, and Immune Escape

As exemplified before, breast microbiotas can have either a protective role against cancer proliferation or an inductive role in tumor cell multiplication [43]. In terms of the influence exerted by microbiotas on cancer metastasis, Fu et al. showed that malignant cells transported bacteria from the primary tumor to the site of the metastasis, as illustrated in Figure 2. Using a 16s rRNA sequencing technique, various tissues were analyzed: the primary breast tumor, normal breast tissue, normal lung tissue, metastasis-adjacent lung with early micro-metastasis, and lung macro-metastasis. It was observed that the breast tumor microbiota resembled the microbiota found in the metastasis-adjacent lung but was completely different from the one found in the normal breast tissue or in the normal lung tissue. The microbiota discovered in the lung macro-metastasis had characteristics from both breast tumor tissue and normal tissues. Further analysis revealed that aerobic types of bacteria were expanding in the lung metastasis, while the amount of facultative anaerobic microorganisms was diminishing [18].

The impact of microbiotas on drug resistance is of great interest, as this interaction between microbiotas and anticancer drugs modulates the immune system, influences side and adverse effects, and eventually can lead to drug resistance as the patient becomes unresponsive to conventional treatment. Rao Malla et al. reported that Doxorubicin is inactivated by streptomyces WAC04685 via a biochemical mechanism of *deglycosylation* [47,48]. The therapy with selective estrogen receptor modulators (SERMs), Tamoxifen and Raloxifene, used in hormone-receptor-positive breast cancer seemed to modulate the microbiota. It was demonstrated that SERMs exert a harmful effect on different bacteria such as Pseudomonas aeruginosa, Bacillus stearothermophilus, Klebsiella pneumoniae, Enterococcus faecium, and many more [47,49]. On the other hand, malignant cells can develop drug resistance to Tamoxifen, as this process is influenced by the microbiota. Gemcitabine, a chemotherapeutic agent used in many types of cancers (including breast cancer), loses its cytotoxic properties when it interacts with microorganisms from *gammaproteobacteria* class [47,50].

## 4. Microbiota in Healthy and Tumoral Breast Tissue 

As stated before, breast tissue has a specific microbiota and it is no longer considered to be sterile, since the local tissular environment (abundant adiposity, rich vasculature, and lymphatic drainage) promotes bacterial growth [14]. In German et al., fresh frozen mammary tissue cores were obtained from 403 healthy (cancer-free) individuals and analyzed for microbial signatures [51]. In terms of family taxonomic rank, *Acetobacterraceae* (*Proteobacteria phylum*), *Lactobacillaceae* (*Firmicutes phylum*), and *Xanthomonadaceae* (*Proteobacteria phylum*) were the most abundant in normal breast tissue. The healthy mammary tissue is mostly colonized by the *Acetobacter genus* (*Proteobacteria phylum*) and *Liquorilactobacillus genus* (*Firmicutes phylum*) [51,52]. Vitorino et al. showed that healthy breast tissue is populated by the following bacterial genera: *Actinobacteria*, *Lactococcus*, *Streptococcus*, *Prevotella*, and *Sphingomonas* [43].

### 4.1. Changes in Bacterial Taxa

It was observed that the composition of microbiotas in patients with breast cancer can vary depending on the type of tissue invaded by the malignant cells. German et al. studied a cohort consisting of 76 breast cancer patients. The fresh frozen tissue cores from the 76 individuals with breast malignancies incorporated primary tumor tissue cores, normal tissue cores adjacent to the tumor mass, and distant-metastasis tissue cores [51]. 

In the normal breast tissue adjacent to the tumor, in terms of the bacterial family level, *Cyanobacteria* and *Corynebacteriaceae* were dominant compared to that in the healthy breast tissue. The *Lactobacillaceae*, *Acetobacterraceae*, and *Xanthomonadaceae* family showed a diminished presence in the normal breast tissue samples adjacent to the tumor compared to the healthy mammary tissue samples. The normal breast tissue adjacent to the tumor was intensely colonized by the *Ralstonia genus* (*Proteobacteria phylum*) in contrast with the normal breast tissue from cancer-free individuals. The *Acetobacter genus* and *Liquorilactobacillus genus* were significantly less abundant in the samples obtained from the normal tissue adjacent to the tumor in contrast to that in the samples obtained from healthy breast tissue [51,52]. 

In terms of the bacterial family taxonomic rank, the primary tumor tissue cores were enriched in *Staphylococcaceae* and *Corynebacteriaceae* in comparison with normal mammary tissue. *Lactobacillaceae*, *Acetobacterraceae*, and *Xanthomonadaceae* were in remarkably lower concentrations in the primary malignant tumor tissue. The *Acetobacter genus* and *Liquorilactobacillus genus* were almost absent in the majority of the primary tumor tissue samples. On the other hand, the *Ralstonia genus* was detected in strikingly high abundance in most of the tumor tissue cores when compared with the abundance in normal breast tissue cores [51]. 

Hoskinson et al. studied and compared the resident microbiotas of healthy breast tissue from cancer-free individuals with the resident microbiota of tissues collected before (prediagnostic) and after (adjacent normal and tumor tissue) breast cancer diagnosis. They highlighted the existence of a transitional taxonomic signature in the prediagnostic tissue compared to that of the healthy breast tissue. The microbiota found in the prediagnostic tissue was an intermediate compositional trademark, indicating the genesis of dysbiosis in the breast tissue prior to malignancy development. In the prediagnostic tissue were discovered several bacterial taxa with similarly higher abundances as in the adjacent normal or tumor tissue, indicating shifts in bacterial taxa (for instance, in *Bacillaceae*, *Burkholderiaceae*, *Corynebacteriaceae*, *Enterobacteriaceae*, *Streptococcaceae*, *Staphylococcaceae*, and *Xanthobacteriaceae*) [53].

The shifts in bacterial taxa and bacterial abundance and the metabolic reprogramming of the microbiota found in breast tissue are preliminary events in malignancy development [53,54]. In terms of metabolic reprogramming of the breast tissue resident microbiota, a downgrading in glutathione metabolism in the adjacent normal and tumor tissue compared to healthy breast tissue was noticed. Glutathione is an important antioxidant chemical compound found in the intracellular fluid, as it regulates cell differentiation, multiplication and apoptosis, and immune defense. Disruptions in glutathione metabolism are involved in tumor growth, development, progression, and therapeutic response [53,55].

### 4.2. Correlation between Host Transcriptome Profiling and Breast Microbiota in Precancerous Tissue

The host–microbiota interactions in breast cancer were highlighted by the correlations found between the host transcriptome and microbial taxa and genes. It was discovered that there was an inverse (negative) host–microbiota correlation pattern between a group of prediagnostic tissues and healthy breast tissues. Most of the correlations between microbial taxa and host transcriptomes and microbial KOs (KEGG orthologs) and host transcriptomes were positive in the prediagnostic tissue, but in the healthy breast tissue, most of these correlations were negative [53,56]. For instance, the CYP24A1 gene, a protein-encoding gene for the 24-hydroxylase enzyme, was one of the host genes positively correlated with microbial genes in prediagnostic tissue and negatively (inversely) linked with microbial genes in healthy mammary tissue [53]. The 24-hydroxylase enzyme is part of the cytochrome P450 family of enzymes [53,57]. These enzymes take part in the metabolic pathway of steroid hormones and xenobiotic substances (drugs, carcinogens, pollutants, and additives) [53,57]. It was observed that cytochrome p40 genes manifest an increased expression in breast malignancies, as we can notice a correlated bacterial response to a transforming tissue microenvironment in the prediagnostic tissue in the incipient stages of breast cancer development [53,57]. All in all, an inverse association between host gene expression and functions of the microbiota can be highlighted [53].

## 5. Microbial Patterns in Breast Cancer

As stated before, the composition of the breast tissue resident microbiota changes dramatically as the breast tissue begins to suffer malignant transformation. Healthy breast tissue hosts mainly three bacterial families: *Acetobacterraceae*, *Lactobacillaceae*, and *Xanthomonadaceae*. The precancerous and cancerous breast tissues (adjacent normal and tumor tissue) are colonized by *Cyanobacteria*, *Corynebacteriaceae*, and *Staphylococcaceae*, in terms of bacterial family taxonomic rank. Breast tissue collected from cancer-free individuals is populated by the *Acetobacter genus* and *Liquorilactobacillus genus*. On the other hand, the normal breast tissue adjacent to the tumor and the tumor tissue are inhabited by the *Ralstonia genus*. The microbiota of the breast tissue collected in a prediagnostic stage acquires a transitional taxonomic signature as shifts in bacterial taxa are made (e.g., *Bacillaceae, Burkholderiaceae*, *Corynebacteriaceae*, *Enterobacteriaceae*, *Streptococcaceae*, *Staphylococcaceae*, and *Xanthobacteriaceae*) [51,52,53] (Table 1).

In a study performed by Kartti et al., 94 fresh samples of tumor and adjacent normal tissue were collected from 47 patients [58]. The researchers investigated the breast cancer tissue samples by the four main molecular subtypes of breast cancer: luminal A, luminal B, HER2-overexpression, and triple-negative breast cancer [5,58]. The composition of the breast microbiota was investigated in tumor tissue and adjacent normal tissue. The main phyla in the two tissue groups are *Proteobacteria* (*class Gammaproteobacteria*), *Firmicutes* (*class Bacilli*), and *Actinobacteria* (*class Actinobacteria*) [15,20,58]. The normal breast tissue adjacent to the tumor was colonized in a higher proportion by microorganisms from the *Gammaproteobacteria class* (37.5%). In the tumor tissue, bacteria from the *Bacilli class* (18.8%) and *Actinobacteria class* (17.2%) were most abundant. It was found that in terms of family taxonomic rank, *Moraxellaceae* prevailed over other bacterial families (*Micrococcaceae*, *Enterobacteriaceae*, and *Staphylococcaceae*) in tumor tissue (19.67%) and adjacent normal tissue (22.32%). *Psychrobacter, Streptococcus, Acinetobacter*, and *Corynebacterium* were the main bacterial genera discovered in the tumor tissue of more than 80% of the individuals selected for this study. The *Streptococcus, Rothia*, and *Staphylococcus genera* were detected in a much higher proportion in tumor tissue compared to that in adjacent normal tissue (which was enriched with the *Escherichia-shigella genus*) [58].

The four main molecular subtypes of breast cancer were examined in terms of resident microbiota diversity. The *Genus Alloiococcus* was dominant in tumor tissue of luminal B subtype. In luminal A subtype, the genus Corynebacterium was plentiful in tumor tissue, while the *Lawsonella genus* colonized the adjacent normal tissue in a higher proportion. The *Sporosarcina genus* is more abundant in the adjacent normal tissue in the triple-negative breast cancer subtype (TNBC). Another disclosure related to TNBC is that the *Sphingomonadaceae family* was the main component of the microbiota in this type of breast malignancy [58]. Breast cancer tissue with HER2 overexpression is mostly colonized by the *Thermus genus*, which includes thermophilic bacteria [58,59].

## 6. Microbiota as a Potential Diagnostic and Prognostic Biomarker

The potential of breast microbiota as a biomarker can be exploited during the various stages of breast cancer development and management: before starting the treatment, in order to establish the molecular profile of the breast malignant tumor; in the course of the treatment, allowing changes in therapy when it becomes ineffective; and during disease progression, in order to determine the gained drug resistance of the tumor [60,61]. 

As a pretreatment biomarker, the breast microbiota would provide valuable information about tumor behavior and tumor response to different antineoplastic therapies. In order to include a biomarker into clinical practice, five stages should be reached: preclinical studies, clinical trials, retrospective, prospective, and control studies [60,62]. At the present time, studies concerning breast microbiotas as biomarkers are still in the preclinical stage; therefore, further research is required in this field [60].

Banerjee et al. analyzed the correlation between microbial signatures of each breast cancer subtype and clinical outcome. This study used 95–105 formalin-fixed paraffin-embedded samples for each breast cancer subtype, 20 paired control specimens, and 68 non-matched control samples obtained from breast reduction interventions [17].

The pan-pathogen microarray (PathoChip) was used to analyze tissue samples (tumor and control samples), highlighting a correlation between distinctive microbial colonization in different breast cancer subtypes and disease prognosis [17,63]. In triple-negative breast cancer tissue samples, higher average hybridization signals of various microorganisms, *Bacillus*, *Mucor*, *Toxocara*, *Trichophyton*, and *Nodaviridae*, could be seen that correlated with prolonged survival time. In the samples collected from “ER + breast cancer” (ER+ and/or PR+ and HER2 -) patients, the analysis performed showed higher average hybridization signals for *Klebsiella, Stenotrophomonas*, and *Neodiplostomum*. This finding was also correlated with extended disease-free time after treatment. For “triple positive” (ER+, PR+ and HER2 +) breast cancer tumor tissue samples, the examination revealed higher average hybridization signals for *Orientia*, *Klebsiella*, *Fusobacterium*, *Azorhizobium*, *Yersinia*, *Arthroderma* (which are all bacterial genera), *Anelloviridae* (a family of viruses), *Angiostrongylus*, and *Toxocara* (parasitic genera). The presence of these microorganisms in “triple positive” breast cancer tissue was strongly correlated with reduced disease-free time after treatment and reduced survival interval. In terms of HER2-overexpression breast cancer, an important correlation between a higher detection of certain microorganisms and a favorable disease prognosis was not found. Nevertheless, some data show that lower average hybridization signals for *Pseudoterranova*, *Trichuris*, *Ancylostoma*, and *Issatchenkia* were correlated with prolonged disease-free period after anticancer therapy [17,64].

Tzeng et al. performed a cross-sectional study which incorporated fresh-frozen breast tissue samples from 221 patients diagnosed with breast cancer and 87 patients without breast malignancy [65]. They described correlations between specific microbial signatures and prognostic features in breast cancer, such as stage, histologic grade, receptor status, and lymphovascular invasion [64,65,66,67,68]. It was found that *Porphyromonas*, *Lacibacter*, *Fusobacterium*, and *Ezakiella* genera were detected in a greater proportion in advanced stage tumors compared to lower stage tumors. They also identified correlations between specific bacterial taxa and markers of tumor metastatic potential. For instance, lymphovascular invasion and node-positive status in breast cancer were associated with a decreased presence of the *Oblitimonas genus*. Furthermore, lymphovascular invasion was positively correlated with the presence of the Lactobacillus genus and negatively correlated with the existence of the *Alkanindiges genus*. A node-positive status in breast cancer was positively correlated with the residence of bacteria from *Acinetobacter* and *Bacteroides genera* and negatively correlated with the presence of microorganisms from the *Achromobacter genus* [65] (Table 2).

Microbiotas from breast tumoral tissue have the potential to complement existing diagnostic and therapeutic methods; however, current data, although promising, do not offer sufficient evidence with regard to the reliability of this method. As such, the application of the routine clinical use of microbiotas as biomarkers for breast cancer requires further research.

## 7. Emerging Role of Microbiota as Therapeutic Target

There is emerging evidence that breast cancer microbiotas can be modulated by antitumor therapy. In certain situations, patients diagnosed with breast cancer receive neoadjuvant chemotherapy (usually a combination of anthracycline, alkylating agents, and taxanes) in order to reduce the size of the tumor before surgery, so that breast-conserving interventions and limited axillary lymph node dissections are feasible [69,70,71]. Chiba et al. investigated if neoadjuvant chemotherapy modulates the tumor-resident microbiota in breast cancer. They collected snap-frozen tumor tissue from patients who received neoadjuvant chemotherapy and from patients who did not undergo prior treatment at the time of surgery. The researchers found that the administration of neoadjuvant chemotherapy diminishes the diversity of microbiota within tumor tissue. The analysis in terms of genus-level differences showed an increased abundance of *Pseudomonas* and a reduced abundance of *Prevotella* in tumor tissue collected from patients who underwent neoadjuvant chemotherapy [69].

Local microbiotas also influence the pathogenesis of hormone-receptor-positive breast cancer in certain ways, by modulating the TME and interfering with the malignant-cell-inner functions [72,73]. These events have an impact as well on the therapeutic efficacy of aromatase inhibitors and estrogen receptor antagonists used in the treatment of hormone-receptor-positive breast cancer [72,74]. As stated before, estrogen receptor antagonists exert toxic effects on various microorganisms related to breast cancer (*Porphyromonas gingivaliis*, *Pseudomonas aeruginosa*, *Klebsiella pneumoniae*, *Acinetobacter baumannii*, *Streptococcus mutants*, *Bacillus stearothemphilus*, and *Enterococcus faecium*) [47,49].

Immunotherapy has become an important constituent in the treatment protocol for many specific cancers, such as melanoma, renal cell carcinoma, non-small-cell lung cancer, gastric cancer, head and neck cancers, and even breast cancer [75].

The use of immune checkpoint inhibitors (ICIs) in the treatment of breast malignancies showed better results compared to conventional chemotherapy, in several clinical trials [76,77,78,79,80,81,82]. Immune checkpoint inhibitors have effectiveness mainly in the triple-negative breast cancer subtype. A phase II clinical trial (KEYNOTE-086) in which Pembrolizumab was used as a frontline treatment for metastatic TNBC revealed an overall response rate of 23%, a good safety profile, and antitumor effectiveness [83,84]. Currently, Pembrolizumab is used in HR (hormone receptor)-negative and HER2-negative breast cancers as a peri-operative therapy in association with neoadjuvant chemotherapy and as an adjuvant systemic therapy, in particular cases, with notable results [85,86,87,88].

It has been discovered that the gut microbiota can influence the potency and toxicity of anticancer treatments, including ICIs. In studies using murine models, a correlation was observed between the existence of certain bacteria within the gut microbiota and a positive response to immunotherapy. A more favorable response to PD-(L)1 inhibitors was seen in murine models colonized by certain bacterial species: *Akkermansia muciniphila*, *Collinsella aerofaciens*, *Bifidobacterium longem*, and *Faecalibacterium prausnitzii* [84,89,90,91,92]. 

Local breast tissue microbiotas and gut microbiotas exert great influence on breast cancer therapy (chemotherapy, hormonotherapy, immunotherapy, and radiotherapy) outcome. Also, a bidirectional interaction between malignant cells and anticancer therapies was observed on one hand and the microbiota on the other [93,94,95,96,97,98]. The estrobolome, which represents the bacterial genes whose products are involved in estrogen metabolism, may increase the risk of breast cancer and may fulfil the role of a biomarker or therapeutic target in the future [99,100,101,102].

## 8. Conclusions

Exploring microbiomes and microbiotas in the context of breast cancer is still a challenging task, but of tremendous importance. It was demonstrated that microbiotas have multiple roles, such as defining healthy and malignant transformed breast tissue, modulating the TME and TIME, or influencing the efficacy of anticancer therapies. There is a strong urge to elucidate the capacity of microbiotas as a large-scale diagnostic and prognostic biomarker in breast cancer. So far, important steps have been taken in this field, analyzing the multifaceted traits of the microbiota in breast malignancies. Still, studies implying the functions of microbiomes and microbiotas deserve more investigation, reflection, and awareness. 

All in all, microbiotas have a great impact on every aspect regarding cancer, exerting influence on the TME, TIME, cancer growth, metastasis, and impacting the outcome of antitumor therapy.

## Figures and Tables

**Figure 1 cancers-16-03468-f001:**
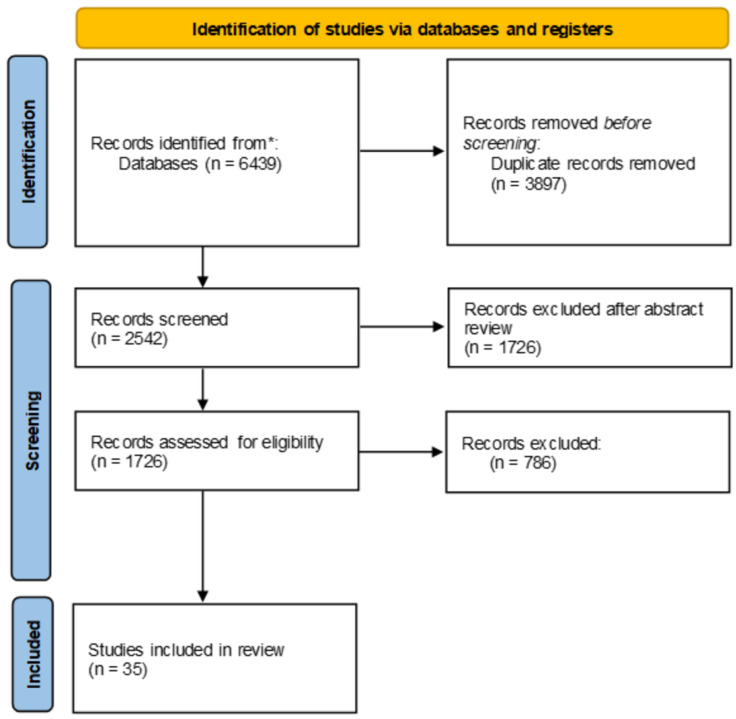
PRISMA flow diagram of article selection process.

**Figure 2 cancers-16-03468-f002:**
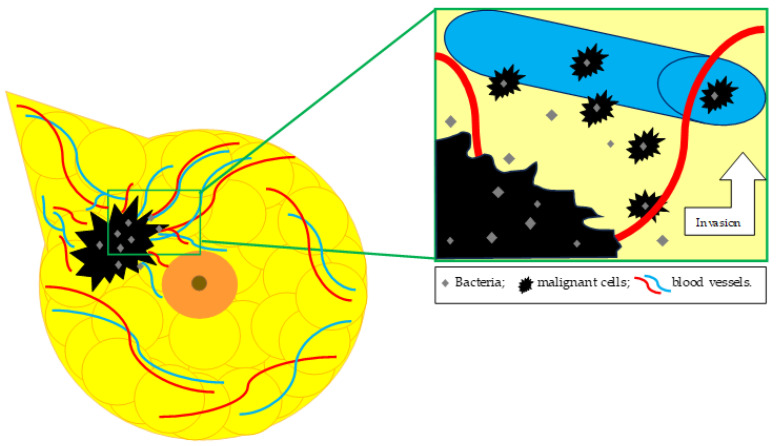
Breast tumoral tissue microbiota in relation to metastatic spread.

**Table 1 cancers-16-03468-t001:** Comparison between the composition of resident microbiota in normal breast tissue, normal breast tissue adjacent to the tumor, and tumor tissue.

Normal Breast Tissue Microbiota	Normal Breast Tissue Adjacent to the Tumor	Breast Tumor Tissue
*Fam. Acetobacterraceae ^1^* *Fam. Lactobacillaceae ^1^* *Fam. Xanthomonadaceae ^1^* *Genus Acetobacter ^1^* *Genus Liquorilactobacillus ^1^* *Genus Actinobacteria ^2^* *Genus Lactococcus ^2^* *Genus Streptococcus ^2^* *Genus Prevotella ^2^* *Genus Sphingomonas ^2^*	*Class Gammaproteobacteria ^3^* *Fam. Cyanobacteria ^1^* *Fam. Corynebacteriaceae ^1^* *Genus Ralstonia ^1^* *Genus Escherichia-shigella ^3^*	*Class Bacilli ^3^* *Class Actinobacteria ^3^* *Fam. Staphylococcaceae ^1^* *Fam. Corynebacteriaceae ^1^* *Fam. Moraxellaceae ^3^* *Fam. Micrococcaceae ^3^* *Fam. Enterobacteriaceae ^3^* *Fam. Staphylococcaceae ^3^* *Genus Ralstonia ^1^* *Genus Psychrobacter ^3^* *Genus Streptococcus ^3^* *Genus Acinetobacter ^3^* *Genus Corynebacterium ^3^* *Genus Rothia ^3^* *Genus Staphylococcus ^3^*

1–German et al. [51], 2023; 2–Vitorino et al., 2022 [43]; 3–Kartti et al., 2023 [58].

**Table 2 cancers-16-03468-t002:** Correlations between tumoral tissue microbiota and BC prognosis.

Author	Publication Year	Type of BC	Tumoral Tissue Microbiota	Observed Association with BC Prognosis
Banerjee et al. [17].		TNBC	*Bacillus*, *Mucor*, *Toxocara*, *Trichophyton* and *Nodaviridae*	High colonization associated with prolonged survival time
	2021	“ER+ BC” (ER+ and/or PR+ and HER2 −)	*Klebsiella*, *Stenotrophomonas*, *Neodiplostomum*	High colonization associated with prolonged survival time
		“Triple positive BC” (ER+, PR+ and HER2 +) HER2-overexpression	*Orientia*, *Klebsiella*, *Fusobacterium*, *Azorhizobium*, *Yersinia*, *Arthroderma*, *Anelloviridae*, *Angiostrongylus*, *Toxocara* *Pseudoterranova*, *Trichuris*, *Ancylostoma*, *Issatchenkia*	High colonization associated with reduced survival time Low colonization associated with reduced survival time
German et al. [51]	2023	NA	*Fam. Staphylococcaceae* and *Corynebacteriaceae* *Genus Ralstonia*	NA
Kartti et al. [58]		Luminal A	*Genus Corynebacterium*	NA
	2023	Luminal B	*Genus Alloiococcus*	NA
		TNBC	*Fam. Sphingomonadaceae*	NA
		HER2-overexpression	*Genus Thermus*	NA
Tzeng et al. [65]	2021	NA	*Porphyromonas*, *Lacibacter*, *Fusobacterium*, *Ezakiella*	High colonization associated with advanced stage tumors
		NA	*Oblitimonas*	Low colonization associated with lymphovascular invasion and node-positive status

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
