# Peer review of "The Influence of Microbiota on Breast Cancer: A Review"

_cancers, 2024, doi:10.3390/cancers16203468_

Round 1

Reviewer 1 Report

Comments and Suggestions for Authors

1)      I strongly suggest the author inserts a graphical abstract at the end of the introduction. This helps the audience get the information faster.

2)      Name of the bacteria in the text must be italic. E.g. S. yanoikuyae in line 93 and M. radiotolerans in line 94.

3)      The authors could use a Venn diagram or other figure type for section 3.3.  (How microbiota is related to cancer proliferation, invasion…)

4)      I suggest the author draws a table for section 5 (microbial patterns in healthy and breast cancer persons)

5)      Regarding microbiota as a prognostic biomarker, how much a microbiota is reliable as biomarker since the biomarker ( microbiota) should be specific. 

Author Response

Reviewer 1

1)      I strongly suggest the author inserts a graphical abstract at the end of the introduction. This helps the audience get the information faster.

R: Thank you for your suggestion. We have added a graphical abstract.

2)      Name of the bacteria in the text must be italic. E.g. S. yanoikuyae in line 93 and M. radiotolerans in line 94.

R: Thank you for your observation, we have corrected this aspect.

3)      The authors could use a Venn diagram or other figure type for section 3.3.  (How microbiota is related to cancer proliferation, invasion…)

R: Thank you for your kind recommendation. We have added a figure for section 3.3.

4)      I suggest the author draws a table for section 5 (microbial patterns in healthy and breast cancer persons)

 R: Thank you for your suggestion. We have transformed the initial figure 2 into a table, as to reduce redundancy with respect to the representation of the microbial patterns in healthy and malignant breast tissue.

5)      Regarding microbiota as a prognostic biomarker, how much a microbiota is reliable as biomarker since the biomarker ( microbiota) should be specific. 

R: Thank you for this observation. Reliability of biomarkers is an essential aspect in the perspective of clinical use. Current evidence suggests that breast tissue microbiota has the potential to become a biomarker, but most available studies are just exploring the correlation with various factors pertaining to breast cancer characteristics and treatment outcomes. As such, at the present time clinical application of microbiota as a biomarker is not yet ready for routine use. We have added a comment with regard to this aspect.

Reviewer 2 Report

Comments and Suggestions for Authors

I believe this paper requires revisions for several reasons. Firstly, the first figure is too simplistic and does not seem appropriate as the initial figure in the paper. A more striking and impactful figure would better capture the reader's attention and set the tone for the study. Additionally,to improve the paper, I recommend adding more detailed analyses and summarizing key findings using either figures or tables. This would provide clearer evidence to support the claims and enhance the overall quality of the paper. Finally, the amount of reference is not enough 

Comments on the Quality of English Language

good

Author Response

I believe this paper requires revisions for several reasons. Firstly, the first figure is too simplistic and does not seem appropriate as the initial figure in the paper. A more striking and impactful figure would better capture the reader's attention and set the tone for the study.

R: Thank you for your recommendation. We have added a graphical abstract and a new figure with more details, that will hopefully be more representative for the scope of this review.

Additionally,to improve the paper, I recommend adding more detailed analyses and summarizing key findings using either figures or tables. This would provide clearer evidence to support the claims and enhance the overall quality of the paper.

R: Thank you for your kind observation. We have revised our paper and added a list of key findings and a further table, along with some additional details in the body of the text.

Finally, the amount of reference is not enough 

R: Thank you for your observation. We have reviewed our selection of papers and have added a few more references that fall within the scope of this review.

Round 2

Reviewer 1 Report

Comments and Suggestions for Authors

The manuscript has been improved. The authors must add the graphical abstract at the end of the introduction. The rest is ok.